

# Wave-optics uncertainty propagation and regression-based bias model in GNSS radio occultation bending angle retrievals

Michael Gorbunov[1] and Gottfried Kirchengast[2,3]

[1]A. M. Obukhov Institute of Atmospheric Physics, Russian Academy of Sciences, Moscow, Russia.
[2]Wegener Center for Climate and Global Change (WEGC), University of Graz, Graz, Austria.
[3]Institute for Geophysics, Astrophysics, and Meteorology/Institute of Physics, University of Graz, Graz, Austria.

*Correspondence to:* Michael Gorbunov (gorbunov@ifaran.ru)

**Abstract.** A new reference occultation processing system (rOPS) will include a Global Navigation Satellite System (GNSS) radio occultation (RO) retrieval chain with integrated uncertainty propagation. In this paper, we focus on wave-optics bending angle retrieval in the lower troposphere and introduce 1. an empirically estimated boundary layer bias (BLB) model then employed to reduce the systematic uncertainty of excess phases and bending angles in the lowest about two kilometers of the

troposphere, and 2. the estimation of (residual) systematic uncertainties and their propagation together with random uncertainties from excess phase to bending angle profiles. Our BLB model describes the estimated bias of the excess phase transferred from the estimated bias of the bending angle, for which the model is built, informed by analyzing refractivity fluctuation statistics shown to induce such biases. The model is derived from regression analysis using a large ensemble of Constellation Observing System for Meteorology, Ionosphere, and Climate (COSMIC) RO observations and concurrent European Centre

for Medium-Range Weather Forecasts (ECMWF) analysis fields. It is formulated in terms predictors and adaptive functions (powers and cross-products of predictors), where we use six main predictors derived from observations: impact altitude, latitude, bending angle and its standard deviation, canonical transform amplitude and its fluctuation index. Based on an ensemble of test days, independent of the days of data used for the regression analysis to establish the BLB model, we find the model very effective for bias reduction, capable of reducing bending angle and corresponding refractivity biases by about a factor of

five. The estimated residual systematic uncertainty, after the BLB profile subtraction, is lower-bounded by the uncertainty from (indirect) use of ECMWF analysis fields but is significantly lower than the systematic uncertainty without BLB correction. The systematic and random uncertainties are propagated from excess phase to bending angle profiles, using a perturbation approach and the wave-optical method recently introduced by Gorbunov and Kirchengast (2015), starting with estimated excess phase uncertainties. The results are encouraging that this uncertainty propagation approach combined with BLB correction enables a

robust reduction and quantification of the uncertainties of excess phases and bending angles in the lower troposphere.

## 1 Introduction

The bending angle and atmospheric profiles retrieval chain for Global Navigation Satellite System (GNSS) radio occultation (RO) data includes many steps involving linear and (moderately) non-linear transformations, starting from excess phase and amplitude measurements (Gorbunov et al., 2006). Error or uncertainty propagation through the geometric optical part of the





retrieval chain has been investigated in a series of theoretical and empirical studies (Kursinski et al., 1997; Syndergaard, 1999; Palmer et al., 2000; Rieder and Kirchengast, 2001; Kuo et al., 2004; Steiner and Kirchengast, 2005; Schreiner et al., 2007; Scherllin-Pirscher et al., 2011b, a, 2017; Innerkofler et al., 2016; Schwarz et al., 2016, 2017a, b; Li et al., 2016, 2017).

The uncertainty propagation through the wave-optical bending angle retrieval block was investigated recently for large-scale
(systematic) and small-scale (random) uncertainties by Gorbunov and Kirchengast (2015), including simulation results demonstrating random uncertainty propagation. Such wave-optical retrieval is essential in the lower troposphere (altitudes below 5 km), where the RO observations are subject to several specific uncertainties not present higher up in the atmosphere, including effects from low signal-to-noise ratio, multipath propagation, and super-refraction (Sokolovskiy, 2001, 2003; Xie et al., 2006; Ao, 2007; Xie et al., 2010; Sokolovskiy et al., 2010).

A thorough treatment of systematic uncertainty and its propagation from excess phase to bending angle in the lower troposphere, including the aim to correct for the known boundary layer bias (BLB) in standard lower troposphere RO retrievals, often termed "negative refractivity bias" (Sokolovskiy et al., 2010; Gorbunov et al., 2015), is lacking so far. Also the propagation of both estimated systematic and estimated random uncertainties through the wave-optical chain, complementary to the geometric-optical uncertainty propagation work of Schwarz et al. (2016, 2017b), was not yet investigated and demonstrated.
This study focuses on providing these missing investigations and on demonstrating BLB correction for a representative large ensemble of real RO data from the COSMIC mission as well as introducing a complete uncertainty propagation approach. The findings and algorithms obtained are used in the development of the new reference occultation processing system (rOPS) including an RO retrieval chain with integrated uncertainty propagation (Kirchengast et al., 2015, 2016a, b).

Our starting points for the BLB model construction are the approach based on refractivity fluctuations introduced by
Gorbunov et al. (2015) and the recent study of RO systematic errors by Gorbunov (2014). Refractivity fluctuations constitute an external factor that results in a systematic shift of the signal phase due to its physical nature rather than any deficiency of the processing algorithm. The strength of this effect can be estimated from the objective characteristics of the signal received. These objective characteristics will hereafter be referred to as predictors in the BLB model. In particular, it was shown already by Gorbunov (2014) that bending angle can serve as such a predictor. Further predictors and the complete BLB model setup
based on a regression-modeling approach are described in this study.

This approach results in the BLB and (residual) systematic uncertainty model formulated in terms of tropospheric bending angles. In order to incorporate this uncertainty modeling into the RO retrieval chain with integrated uncertainty propagation, it needs to be transferred into the equivalent excess phase BLB and (residual) systematic uncertainty estimate. For its propagation then a perturbation approach or the approximation derived by Gorbunov and Kirchengast (2015) can be employed. In that
paper we discussed the propagation of excess phase to bending angle uncertainty through the Fourier Integral Operator (FIO) used for the bending angle retrieval (Gorbunov and Lauritsen, 2004). This uncertainty propagation uses the stationary phase approximation, which allowed for the derivation of simple propagation formulae.

In order to now transform the bending angle uncertainty into the equivalent excess phase uncertainty, we use the inverse FIO, which was recently employed by Gorbunov (2016) for the retrieval of reflected rays from RO data. Specifically, the systematic
uncertainty is evaluated for every RO event in the form of estimated profiles of bending angle BLB and (residual) systematic



uncertainty. These estimates are then transformed into the equivalent BLB and (residual) systematic uncertainty of the excess phase, where they complement the estimated random and basic systematic uncertainty of the excess phase, available separately from the preceding step of excess phase processing (Innerkofler et al., 2016; Schwarz et al., 2016, 2017b). Both together are used as input to the wave-optical uncertainty propagation.

The paper is organized as follows. In Sect. 2 we describe the empirical BLB model, based on a regression analysis guided by the understanding that refractivity fluctuation statistics induce such biases, as well as a simple (residual) systematic uncertainty model for the BLB-corrected bending angles. Section 3 describes the wave-optical propagation of estimated systematic and random uncertainties from excess phase to bending angle, for the methodology also recalling the key results needed from Gorbunov and Kirchengast (2015) and Gorbunov (2016). In Sect. 4 we discuss the results of the application of the BLB correc-

tion based on a large ensemble of COSMIC RO data from representative test days throughout the year 2008. Section 5 provides our conclusions.

## 2   Boundary Layer Bias (BLB) Model of Bending Angle and its Uncertainty

The BLB model is formulated to be capable of providing bending angle BLB profiles over the lower troposphere up to 5 km impact altitude, corresponding to about 4 km (mean-sea-level) altitude, with the primary bias effects occuring within the atmo-

spheric boundary layer below about 2 km altitude. Here we describe its setup by first introducing the underlying refractivity fluctuations model (Sect. 2.1) then followed by the BLB model description (Sect. 2.2). The model is built as a regression model using adaptive functions based on predictors available for each RO event, including impact altitude, latitude, bending angle (BA), BA standard deviation, canonical transform (CT) amplitude, and CT fluctuation index as main ones. The selection of the predictors is explained in Sect. 2.3 and their use in constructing the adaptive functions in Sect. 2.4.

Along with the decription we illustrate the performance of the BLB model to quantify the boundary layer biases based on the predictors, underpinning that the BLB profiles obtained for individual RO events can be effectively used for BLB correction and lead to a significant reduction of systematic uncertainty. A simple model for the estimated residual systematic uncertainty after the BLB profile subtraction, which is accounting for the residual bias and the uncertainty (indirectly) incurred from the use of ECMWF analysis profiles as regression reference, is desribed in Sect. 2.5.

### 2.1   Underlying Model of Refractivity Fluctuations

In order to understand the bending angle BLB in terms of "negative refractivity bias" (Sokolovskiy et al., 2010; Gorbunov et al., 2015) we use the fluctuation-based model introduced by Gorbunov et al. (2015). Illustrating this modeling approach, Figure 1 shows an example profile of the refractivity structure constant $C_N^2(z)$ and the corresponding relative difference statistics of an ensemble of bending angle and refractivity profiles. The latter were obtained by comparison of the modeled "truth"

based on ECMWF refractivity fields, used as reference, and perturbed data based on the same ECMWF fields but with random refractivity fluctuations according to the $C_N^2(z)$ profile superimposed. The $C_N^2(z)$ profile was tuned to realistically represent





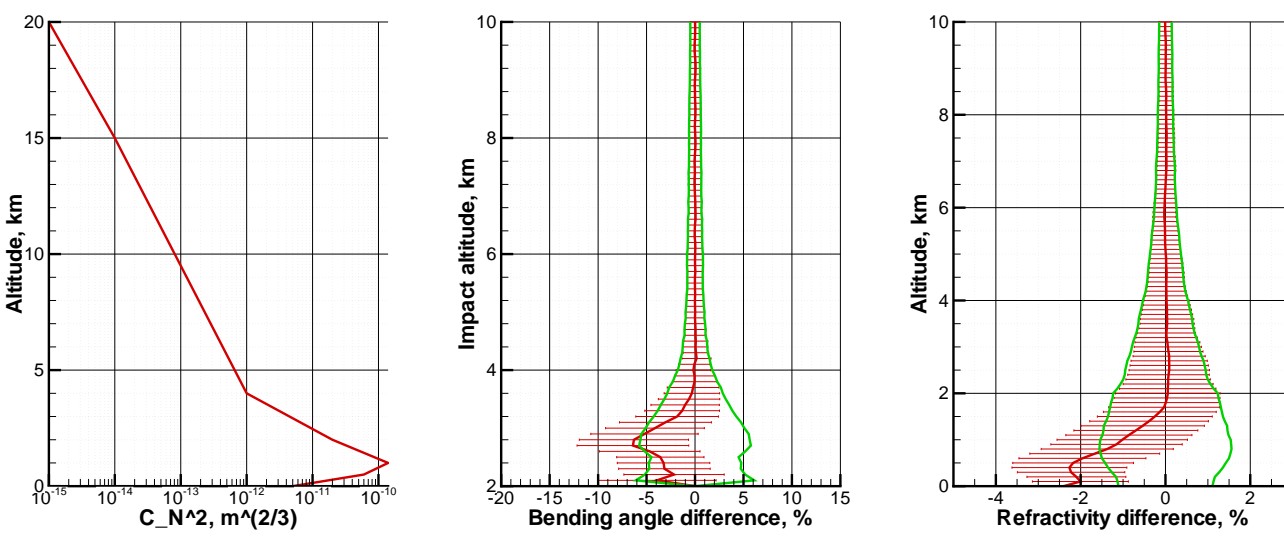

**Figure 1.** Deviation statistics induced by simulated refractivity fluctuations: refractivity structure constant $C_N^2(z)$ profile (left) and associated difference statistics of ECMWF profiles with and without fluctuations superposed, for bending angle as function of impact altitude (middle) and refractivity as function of altitude (right), where mean difference (red), standard deviation (green) and the difference-ensemble spread (horizontal bars at vertical levels) are shown. COSMIC event geometry and concurrent ECWMF analysis fields from the 15th day of every month of year 2008 were used to produce the statistics.

BLB statistics of RO observations and the wave optics propagator (WOP) package (Gorbunov, 2011) was used to realistically compute the bending angles.

It is visible in Fig. 1 that the refractivity fluctuations lead to a negative refractivity bias of up to about 2 % in the boundary layer and an associated negative BLB in bending angle of up to about 5 %, typical of biases seen in real RO data. Random
5    differences (standard deviation) reach realistic values as well, about 1.5 % in refractivity and about 5 % in bending angle.

To put these simulation results into direct context with real data, Fig. 2 shows another set of difference statistics for bending angles and refractivities, from low latitudes to high latitudes, where we again used the modeled "truth" from ECMWF fields as reference but now to illustrate the differences of observed profiles from COSMIC. These results confirm that refractivity fluctuations can explain and quite well describe the systematic and random error behavior of RO bending angles and refractivities
10    in the boundary layer. A somewhat higher level of RMS deviations (standard deviation) seen for the COSMIC data, compared to Fig. 1, is likely caused by the fact that ECMWF fields themselves deviate from the real atmospheric state (see, e.g., the error modeling of Scherllin-Pirscher et al. (2011b, 2017)).

Based on this understanding we can robustly assume that reliable modeling of the bending angle BLB, and subsequent use of the model for BLB correction, will also effectively mitigate biases in the retrieved refractivity profiles and further-derived
15    atmospheric profiles. However, given the highly variable refractivity fluctuations affecting individual RO events in reality,



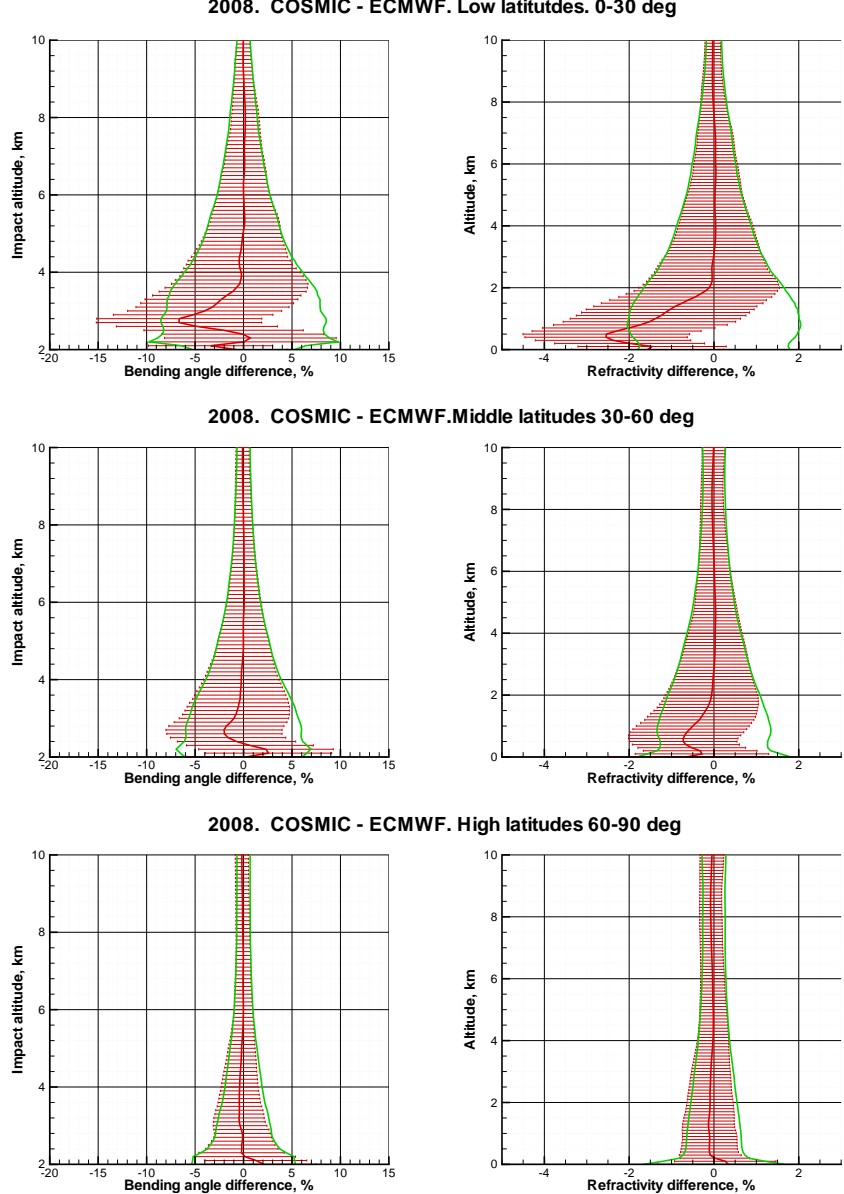

**Figure 2.** Deviation statistics obtained for real RO data: difference statistics of COSMIC profiles including real fluctuations relative to ECWMF profiles without fluctuations, for bending angle as function of impact altitude (left column) and refractivity as function of altitude (right column), with same style of panels as for the difference statistics in Fig. 1. Results for low latitudes (top), mid latitudes (middle), and high latitudes (bottom) are shown, for COSMIC events and concurrent ECWMF analysis fields from the 15th day of every month of year 2008.



which implies a complex dependence of the BLB on the RO location and the data characteristics of the individual RO profiles, we also realize that we need to implement a BLB model with a very flexible functional behavior in order to reliably serve its purpose. We therefore have chosen a highly versatile empirical regression modeling approach described next.

## 2.2 Bending Angle BLB Model from Regression to Adaptive Functions

We model the BLB by a predictor-based empirical model that is flexible enough to capture the BLB behavior by suitable predictors under widely variable predictor value ranges for individual RO events. Because the dependence of the BLB model profiles from predictors is unknown *a priori*, we solve for this dependence in the form of linear combination of a set of linear and non-linear functions of the predictors. We refer to these functions as adaptive functions. The model estimate of the regression coefficients of the linear combination is based on the comparison of a large set of bending angle observations with
a reference data set.

In this study, introducing a first reliable BLB model version, the observations are from the COSMIC mission and the reference data set consists of gridded fields of meteorological variables from ECMWF. The ECMWF data have their own systematic uncertainty, which is taken into account by letting these uncertainties flow into the estimated residual systematic uncertainty of bending angle profiles after BLB correction (Sect. 2.5).

The BLB model is formulated as follows. We used a set of COSMIC bending angle observations, including 24 representative days from year 2008. We adopted the 15th and 16th day of every month, amounting in total to about 54000 RO events. We used the corresponding ECMWF fields as basis for obtaining the "true" reference bending angles. To this end, we employed the Wave Optics Propagator (WOP) (Gorbunov, 2011) to generate the bending angle profiles from the ECMWF refractivity fields. We then performed a regression of the differences of observed and reference bending angles in the lower troposphere with
respect to the chosen adaptive functions (Sect. 2.4). The adaptive functions are formulated in terms of predictors, which are evaluated from objective characteristics of every RO event, without using the reference data (Sect. 2.3). These ingredients allow for the derivation of regression coefficients, which upon their estimation complete the BLB model then ready to be applied based on predictors from a given RO event.

Because we need to derive the regression model for widely diverse BLB behavior as noted above, we start with very general
regression relations. Consider two series of random variables, vector $\boldsymbol{x}_i$ and scalar series $y_i$, where the lower index $i$ enumerates the realizations. We will term the components of $\boldsymbol{x}_i$ predictors, because we approximate the random variables $y_i$ as a linear combination of pre-defined adaptive functions $\varphi^j$ of $\boldsymbol{x}_i$. The number of predictors, and of associated adaptive functions, is much smaller than the number of realizations (difference profiles of observed and reference bending angles in the lower troposphere). We write the over-determined system of equations,

$$y_i = \sum_j \alpha^j \varphi^j (\boldsymbol{x}_i) \equiv \sum_j \alpha^j K_{ij}, \tag{1}$$

$$K_{ij} = \varphi^j (\boldsymbol{x}_i), \tag{2}$$





or in the vector form,

$$\boldsymbol{y} = \hat{K}\boldsymbol{\alpha}. \tag{3}$$

This system has a pseudo-inverse solution, i.e., the vector $\boldsymbol{\alpha}$ that minimizes the discrepancy

$$\left(\boldsymbol{y} - \hat{K}\boldsymbol{\alpha}\right)^T \left(\boldsymbol{y} - \hat{K}\boldsymbol{\alpha}\right) = \min \tag{4}$$

is obtained as the least-squares solution of this overdetermined problem in the form

$$\boldsymbol{\alpha} = \left(\hat{K}^T \hat{K}\right)^{-1} \hat{K}^T \boldsymbol{y}. \tag{5}$$

Now consider a numerical estimation of $\boldsymbol{\alpha}$ that allows for an evaluation readily augmentable in terms of number of realizations and adaptive functions. Preparing the quadratic form

$$\hat{B} = \hat{K}^T \hat{K}, \tag{6}$$

$$B_{ij} = \sum_k K_{ki} K_{kj} = \sum_k K_{ki} K_{kj} = \sum_k \varphi^i(\boldsymbol{x}_k)\varphi^j(\boldsymbol{x}_k), \tag{7}$$

we have available matrix $\hat{B}$ as a square symmetric matrix that can be evaluated by the summation over any existing set of realizations of $\boldsymbol{x}_i$. Similarly, using the transform

$$\boldsymbol{z} = \hat{K}\boldsymbol{y}, \tag{8}$$

$$z_i = \sum_j K_{ij} y_j = \sum_j \varphi^i(\boldsymbol{x}_j) y_j, \tag{9}$$

we have available vector $\boldsymbol{z}$ as a vector that can also be evaluated by the summation over any existing set of realizations of $\boldsymbol{x}_i$ and $y_i$. Finally, it is straightforward in this formulation to obtain the regression coefficients from

$$\boldsymbol{\alpha} = \hat{B}^{-1}\boldsymbol{z}. \tag{10}$$

For convenience, matrix $\hat{B}$ and vector $\boldsymbol{z}$ can be redefined in terms of averaging over the ensemble of realizations. Denoting $N$ the number of realizations, this is performed by dividing both $\hat{B}$ and $\boldsymbol{z}$ by $N$,

$$B_{ij} = \frac{1}{N}\sum_k \varphi^i(\boldsymbol{x}_k)\varphi^j(\boldsymbol{x}_k) = \left\langle \varphi^i \varphi^j \right\rangle, \tag{11}$$

$$z_i = \frac{1}{N}\sum_j \varphi^i(\boldsymbol{x}_j) y_j = \left\langle \varphi^i y \right\rangle. \tag{12}$$

Practically, normalization can also be an issue, depending on the number of adaptive functions. If their number is as high as about 200 such as in our study (Sect. 2.4) then even a small change of the normalization factor is raised to the 200th power when evaluating the matrix determinant. This may result in overflow or underflow in the matrix inversion. Therefore, the numerical

algorithm requires accurate tuning of the normalization factor in order to ensure a stable and robust inversion of matrix $\hat{B}$.

After having solved for the regression coefficient vector $\boldsymbol{\alpha}$ it can be used within Eq. 3, which then serves as the BLB model applicable to any given RO event. It will provide the estimated bending angle BLB profile $\boldsymbol{y}$ for the RO event when its predictors are used to specify the model matrix $\hat{K}$.





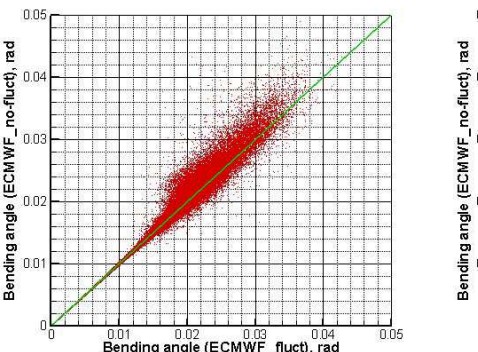 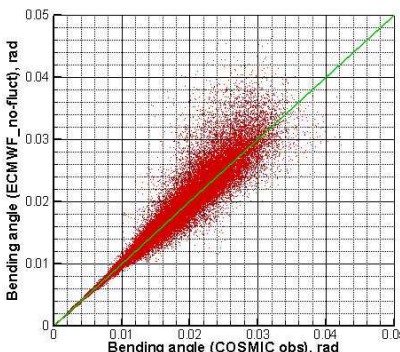

**Figure 3.** Scatter plot of fluctuation-affected bending angle profiles (x-axis) from ECMWF-based simulations with refractivity fluctuations superposed (left) and from COSMIC observations (right), respectively, versus reference bending angle profiles (y-axis) from ECMWF simulations without refractivity fluctuations superposed. COSMIC events and concurrent ECWMF analysis fields from the 15th and 16th day of every month of year 2008 were used for these example results.

## 2.3 Predictors for the Model's Adaptive Functions

Here we consider the predictors that we may reasonably choose. Besides predictors depending on RO event altitude and latitude (discussed separately below) we adopt the following four predictors that are derived from observational RO data, all as function of impact parameter $p$ within the lower troposphere (below an impact altitude of 4.5 km): 1) bending angle $\epsilon(p)$, 2) bending

5    angle standard deviation $\delta\epsilon(p)$, 3) normalized CT amplitude $A_{CT}(p)$, and 4) CT amplitude fluctuation index $\beta(p)$. Bending angle standard deviation is the bending angle standard error estimate based on radio-holographic analysis (Gorbunov et al., 2006). The CT amplitude (Gorbunov, 2002; Gorbunov and Lauritsen, 2004) is the measure of energy density of rays in the impact parameter space. We use the CT amplitude normalized in such a way that it should equal unity in vacuum. The CT amplitude fluctuation index $\beta(p)$ is defined as,

$$\beta(p) = \hat{S}_\beta \left( \left( A_{CT}(p) - \hat{S} A_{CT}(p) \right)^2 \right), \tag{13}$$

where $\hat{S}_\beta$ is a smoothing operator (lowpass filter) for which we use a 2 km smoothing width.

     Figure 3 shows the scatter plot of fluctuation-affected bending angles versus reference bending angles for the fluctuation-model simulations (like for Fig. 1) and the COSMIC observations (like for Fig. 2). In both cases the asymmetry with respect to the diagonal is visible (fluctuation-affected bending angles tentatively smaller than reference ones). This indicates that the

15    bending angle itself can serve as one meaningful predictor of (negative) boundary layer biases.

     Figure 4 shows scatter plots of the difference of fluctuation-affected and reference bending angle profiles versus bending angle standard deviation (top), normalized CT amplitude (middle), and CT amplitude fluctuation index (bottom), for simulations (left) and COSMIC observations (right). Comparing the behavior of these predictors, their correlation with the bending angle difference is clearly more salient in the simulations but some smaller asymmetry can also be noticed for the COSMIC



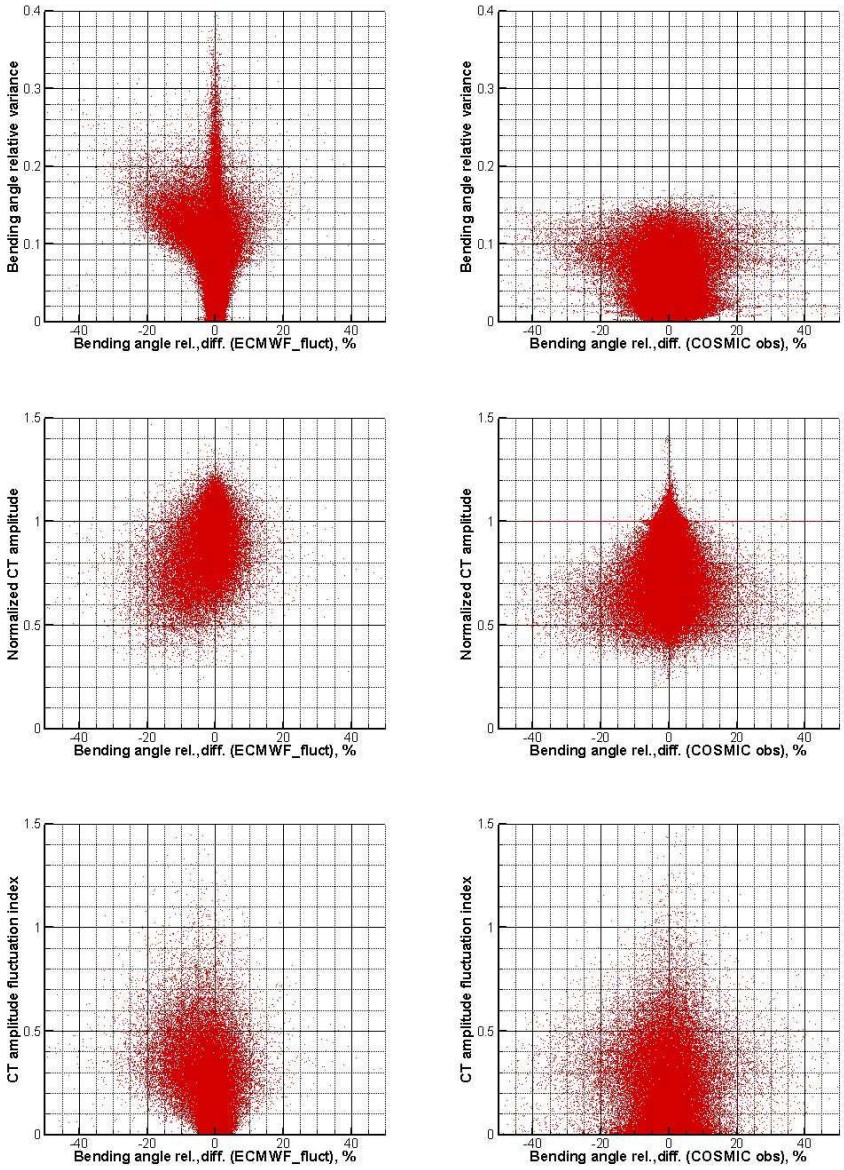

**Figure 4.** Scatter plot of the the difference of fluctuation-effected and reference bending angle profiles (x-axis), for ECMWF simulations with refractivity fluctuations superposed (left column) and COSMIC observations (right column), respectively, versus the predictor variables (y-axis) bending angle standard deviation (top), normalized CT amplitude (middle), and CT amplitude fluctuation index (bottom). The reference bending angles are from ECWMF simulations without refractivity fluctuations superposed. The same ECWMF fields and COSMIC data as for Fig. 3 were used.





observation differences. We therefore kept all four predictors in this study and left possible further reduction of these predictors (and associated adaptive functions) to future fine-tuning of the BLB model regression.

In addition to these four predictors we utilize the RO event coordinates impact altitude $z$ and latitude $\lambda$, where $z = p - R_{LC} - U_{geoid}$, with $R_{LC}$ the local radius of curvature and $U_{geoid}$ the geoid undulation applying to the event location. We use

the impact altitude $z$ directly and in form of the following six trigonometric functions of $z$,

$$\sin\left(2\pi n \frac{z - z_{min}}{z_{max} - z_{min}}\right), \cos\left(2\pi n \frac{z - z_{min}}{z_{max} - z_{min}}\right), n = 1...3, \tag{14}$$

where $z_{min}$ and $z_{max}$ are the limits of impact altitude wherein the BLB profiles are evaluated (equal to 1.5 km and 4.5 km). Latitude $\lambda$ is used in form of another six trigonometric functions of $\lambda$,

$$\sin\left(n\lambda\right), \cos\left(n\lambda\right), n = 1...3. \tag{15}$$

Altogether we therefore use $N_p = 17$ predictors, including impact altitude, the four observation-derived predictors, six functions of impact altitude, and six functions of latitude.

## 2.4 Construction of the Model's Adaptive Functions

General adaptive functions as we use here are constructed in form of different degrees of the predictors and their cross-products, from degree zero, which produces unity, up to some maximum degree $D_p$,

$$\left\{\varphi^j\left(\mathbf{x}\right)\right\} = \left\{1, \quad \left(x^i\right)^\gamma, \quad \left(x^{i_1}\right)^{\gamma_1}\left(x^{i_2}\right)^{\gamma_2}\right\}, \tag{16}$$

$$1 \le i \le N_p, \quad 1 \le \gamma \le D_p, \tag{17}$$

$$1 \le i_1 < i_2 \le N_p, \quad 1 \le \gamma_1 + \gamma_2 \le D_p, \quad \gamma_{1,2} > 0. \tag{18}$$

We use a maximum degree of $D_p = 3$ and apply some additional constraints further limiting the adaptive functions to the reasonable ones. For the six trigonometric functions of impact altitude (Eq. 14) it is not allowed to take their degrees beyond

degree 1 and their cross-products as these will not be linearly independent from other trigonometric functions of the impact altitude. The same applies to the six trigonometric functions of latitude (Eq. 15) for which we therefore also disregard degrees beyond degree 1 and cross-products.

For our choice of $D_p = 3$ we thus obtain $N_f = 214$ adaptive functions. To understand this number, consider different degrees of predictors. There is one 0-degree function (unity). There are 17 functions of degree 1 (the 17 predictors). There are $6 \times (6 +$

$5) + 6 \times 5 + (5 \times 4)/2 + 5 = 111$ functions of degree 2. There are $2 \times 6 \times 5 + 5 + 5 \times 4 = 85$ functions of degree 3. Therefore, we arrive in total at $1 + 17 + 111 + 85 = 214$ adaptive functions, which provide the needed flexibility for the highly variable BLB profile behavior while still allowing for a robust estimation of the regression coefficients. If future fine-tuning of the regression model would reduce the number of predictors, the number of adaptive functions would reduce accordingly.





## 2.5 Simple Residual Systematic Uncertainty Model

As described in Sect. 2.2, after obtaining the regression coefficient vector (Eq. 10) we can use it within the regression model (Eq. 3), which then serves as the BLB model applicable to any given RO event. It provides the bending angle BLB model profile for the RO event, $\delta\alpha_{\mathrm{BLB}}(z)$, based on its predictors depending on location (impact altitude, latitude) and bending angle and CT amplitude characteristics (Sect. 2.3).

Given this basis, we define a simple intial systematic uncertainty model for the BLB-corrected bending angle profiles of the lower troposphere, $u^s_{\delta\alpha,\mathrm{BLB}}(z)$, which consists of two components: 1.) an estimated "lower bound" ECWMF reference field-induced systematic uncertainty, $u^s_{\mathrm{refEC}}$, that accounts for the uncertainty from using the ECMWF analysis fields as the regression reference which have their own (small) systematic deviations from the "truth", and 2.) an estimated residual bias uncertainty after BLB correction by subtracting the BLB model profile, $u^s_{\mathrm{resBLB}}$, since the empirical-statistical BLB regression model can never fully fit the individual bias situation of an RO event.

From experience with estimated biases of ECWMF analysis fields in other studies (e.g., Li et al., 2013, 2015; Scherllin-Pirscher et al., 2017; Li et al., 2017) we formulate the model for the ECMWF reference field-induced systematic uncertainty profile $u^s_{\mathrm{refEC}}(z)$ as a fractional model ($f^{us}_{\mathrm{refEC}}(z)$) with a linear increase downward over the lower troposphere towards the surface,

$$100 \cdot \frac{u^s_{\mathrm{refEC}}(z)}{\alpha_{\mathrm{refEC}}(z)} = f^{us}_{\mathrm{refEC}}(z) = f^{us}_{\mathrm{refEC,zmin}} \cdot \frac{(z_{max} - z)}{(z_{max} - z_{min})}, \tag{19}$$

where $\alpha_{\mathrm{refEC}}(z)$ is the ECMWF reference bending angle profile, $z_{min}$ and $z_{max}$ are the limits of impact altitude (set to 1.5 km and 5.0 km), and $f^{us}_{\mathrm{refEC,zmin}}$ is the fractional uncertainty at $z_{min}$ empirically set to 0.25 %. For perspective, the bending angle uncertainties obtained this way correspond in terms of temperature to uncertainties from about 0.2 K near 4 km impact altitude to 0.6 K near the surface (for details on uncertainty relations among RO-derived variables see Scherllin-Pirscher et al. (2011b) and Scherllin-Pirscher et al. (2017) and references therein).

The estimated residual bias uncertainty profile after BLB correction is formulated from experience with other bias corrections, such as sampling bias correction (e.g., Scherllin-Pirscher et al., 2011a, 2017), and based on BLB correction performance results with test ensembles during this study, in a straightforward fractional form,

$$u^s_{\mathrm{resBLB}}(z) = r_{\mathrm{resBLB}} \cdot \delta\alpha_{\mathrm{BLB}}(z), \tag{20}$$

where $r_{\mathrm{resBLB}}$ is the systematic uncertainty reduction factor empirically set to 0.2, i.e., expressing that due to the BLB correction the bias in the bending angle profile is reduced by a factor of five.

For the estimated residual systematic uncertainty finally attributed to the BLB-corrected lower tropospheric bending angle at any impact altitude we then simply adopt the larger one of the two uncertainties,

$$u^s_{\delta\alpha,\mathrm{BLB}}(z) = \hat{S}_{us}\left(\mathrm{Max}\left(u^s_{\mathrm{resBLB}}(z), u^s_{\mathrm{refEC}}(z)\right)\right), \tag{21}$$

implementing the "lower bound uncertainty" role of $u^s_{\mathrm{refEC}}$ in case the estimated residual bias uncertainty $u^s_{\mathrm{resBLB}}$ of individual RO events according to Eq. 20 is occasionally very small. $\hat{S}_{us}$ is a smoothing operator (lowpass filter) with a 0.4 km filter width





that we use to ensure adequate smoothness of the resulting $u_{\delta\alpha,\mathrm{BLB}}^s(z)$ profile also over those altitude levels where the two uncertainty components cross in their magnitude.

## 3 Wave-Optical Propagation of Systematic and Random Uncertainties

The propagation of systematic and random uncertainties through the wave optical retrieval chain was investigated by
Gorbunov and Kirchengast (2015), where a simple approximation was derived and verified based on numerical simulations (as summarized in Sect. 1). The approximation considers the excess phase as function of time, $\Psi(t)$, and its systematic ("small-scale") and random ("large-scale") uncertainties, $\Sigma_1(t)$ and $\Sigma_2(t)$, respectively. The uncertainty in the impact parameter space (Gorbunov and Lauritsen, 2004) is then evaluated as $\tilde{\Sigma}_{1,2}(p) = \Sigma_{1,2}(t(p))$, where $t(p)$ is the time of observation of the ray with impact parameter $p$.

Practically the application of this approximation was shown by Gorbunov and Kirchengast (2015) to work well for propagating random uncertainties (covariance matrices), while in sensitivity tests and evaluations for this study we found that it does not work sufficiently well for propagating systematic uncertainties, due to the large-scale nature of such (increment) profiles not transforming smoothly under FIO operations (Gorbunov and Lauritsen, 2004). Similarly, given the BLB and residual systematic uncertainty model being formulated in terms of bending angle, their inverse transformation into the equivalent excess
phase bias and uncertainty proves to be not straightforward either.

The reason and underlying problem is that the perturbation of the excess phase due to superimposing the systematic uncertainty of the bending angle is not smooth. The variation of the bending angle profile in each realization results in different phase perturbation corresponding to a different ray manifold with a different caustic structure. Therefore, the excess phase perturbation has a complicated non-linear relation with the phase (eikonal) uncertainty in impact parameter space, and this
perturbation corresponds to a complicated coherent signal being a superposition of multiple signals corresponding to different rays.

To overcome this difficulty, we do apply the linearized approximation only for the propagation of random uncertainty, i.e., the covariance propagation according to Gorbunov and Kirchengast (2015); Eqs. (29) and (30) therein. This is applied within the rOPS wave-optical retrieval, for both GNSS frequencies, right after the bending angle profiles themselves have
been retrieved by the (forward) FIO in CT2 implementation (Gorbunov and Lauritsen, 2004; Gorbunov, 2011). The BLB and estimated systematic uncertainty propagation is then computed, in a consistent way for bending angles and excess phases, with a perturbation approach in a three-step sequence as follows.

First, the BLB profile and its estimated systematic uncertainty profile after BLB subtraction are computed according to Sect. 2.5 for the lower tropospheric bending angle profile at the L1 frequency, for the location and characteristics (i.e., the
applicable predictors) of the given RO event. It is not computed for the second (L2) frequency, since the L2 profiles are generally more noisy (making BLB estimation difficult) and anyway not further used at impact altitudes below 5 km. Below this level, where the neutral atmospheric excess phase always exceeds several hundreds of meters, the dual-frequency ionospheric correction rather always uses L1–L2 difference bending angles extrapolated from above (Schwarz et al., 2017b), avoiding





noise amplification and mitigating potentially adverse effects on top-of-boundary-layer (TBL) estimates recently pointed out by Sokolovskiy et al. (2016).

Second, the BLB-corrected L1 bending angle profile, and this profile perturbed by the estimated systematic uncertainty profile, are each projected back to excess phase by applying the inverse FIO approach recently introduced by Gorbunov (2016).
This provides the BLB-corrected L1 excess phase profile and, from the difference of the two back-projected profiles, the estimated systematic excess phase uncertainty profile pertaining to it. The latter BLB-related systematic uncertainty is then added (in a root-mean-square sense) to the basic systematic excess phase uncertainty available from the raw processing towards excess phase (Innerkofler et al., 2016), yielding the total estimated systematic excess phase uncertainty profile.

Third, the BLB-corrected L1 excess phase profile, and this profile perturbed by the total estimated systematic uncertainty
profile, are processed again through the standard (forward) FIO CT2-wave-optics retrieval in order to obtain a BLB-corrected retrieved bending angle profile, for consistency check with the orginal BLB-corrected bending angle profile, as well as the total estimated systematic bending angle uncertainty profile, from the difference of the two CT2-retrieved bending angle profiles. The systematic bending angle uncertainty profile at the second (F2) frequency is finally obtained from processing also the L2 excess phase profile perturbed by its associated systematic uncertainty through the wave-optics retrieval and estimating it from
the difference of the resulting perturbed bending angle profile to the one originally retrieved from the unperturbed L2 excess phase.

Despite of the complexities from the non-liniearites involved, we obtain in this way a conistent set of excess phase and bending angle profiles together with their estimated systematic and random uncertainties, which are BLB-corrected at the L1 frequency in the lower troposphere. The extra computational expense for the uncertainty propagation due to the non-
linearity is reasonably limited to one additional forward and inverse FIO operation at L1 frequency, required for the perturbation approach to systematic uncertainty propagation. This is similar to the uncertainty propagation work of Schwarz et al. (2017a) and Schwarz et al. (2017b), where the perturbation approach is also needed in a small number of steps (during geometric-optics bending angle retrieval and dry-air temperature retrieval) for the systematic uncertainty propagation.

## 4   Results

Here we evaluate the consistency of the BLB-corrected bending angles and their asssociated retrieved refractivities from either using the original BLB-corrected bending angles directly or from using the BLB-corrected retrieved bending angles, i.e., those from first back-projecting the original bending angles to obtain BLB-corrected excess phases and then retrieving the bending angles again. This provides a basic validation of our procedure as described in Sect. 3; for limiting the extent of this paper the detailed inspection and validation of the uncertainty propagation itself is left for a follow-on study.
We investigated the BLB-correction of an independent ensemble of COSMIC-retrieved bending angles employing our BLB model, as in Sect. 2.1 using ECWMF analysis fields as reference. Figure 5 shows the COSMIC-ECMWF difference statistics of bending angles (left) and refractivities (right) after bending angle BLB correction. These statistics were evaluated for a set of 12 days of COSMIC data from year 2008, including the 17th day of every month, amounting in total to about 26000 RO events.



This implies that these COSMIC and ECMWF ensembles of profiles are independent from the ones used in the derivation of the BLB model regression coefficients (the 15th and 16th day of every month; cf. Sect. 2.1).

Cross-checking these results with results from COSMIC and ECMWF ensembles using the 16th day of every month (not separately shown), we find them practically indistinguishable in terms of their difference statistics. This indicates the statistical

homogeneity of the data sets and the robustness of the BLB model. Furthermore, from comparing Fig. 5 with Fig. 2, it is clear that the BLB correction achieves a substantial decrease of the boundary layer biases, by about a factor of five, consistent with the systematic uncertainty reduction factor $r_{\mathrm{resBLB}} = 0.2$ (Eq. 20). Immediately above the boundary layer, above about 2 km altitude, the BLB-corrected profiles possibly contains slightly increased uncertainty, at small magnitude, which is accounted for by the reference field-induced "lower bound" uncertainty $u^s_{\mathrm{refEC}}$ (Eq. 19) included in the systematic uncertainty model up

to 5 km impact altitude. This may be improved in the future by further refined BLB model design.

Figure 6 shows the COSMIC-ECMWF difference statistics of bending angles (left) and refractivities (right) based on the BLB-corrected retrieved bending angles, i.e., those from first back-projecting the original bending angles by the inverse FIO to obtain BLB-corrected excess phases and then retrieving the bending angles again. Except for about the lowest half-kilometer above surface where there is possibly some degradation, the results are found very close to those shown in Fig. 5 for the original

BLB-corrected bending angles. This indicates the basic validity and robustness of our approach to transfer the BLB-corrected bending angles to BLB-corrected excess phases (and via perturbation approach also the associated systematic uncertainties). Future more detailed inspection of the full uncertainty propagation approach according to Sect. 3 will consolidate this encouraging initial validation.

## 5  Conclusions

In this study we developed a regression-based approach for modeling and propagating atmospheric boundary layer biases (BLBs) and associated (residual) systematic uncertainties within the wave-optical retrieval chain of the reference occultation processing system (rOPS), a new RO processing system with integrated uncertainty propagaqtion that focuses on calibration/validation and climate applications.

The starting point encouraging and informing our BLB model design was fluctuation-based explanatory modeling of the

well known "negative refractivity bias" problem in the boundary layer. We showed that it is possible to achieve a reasonable agreement with observed bending angle and refractivity biases by modeling fluctuation statistics consistent with reasonable tropospheric profiles of the refractivity structure constant $C^2_N(z)$.

Based on this understanding we can robustly assume that reliable modeling of the bending angle BLB, and subsequent use of the model for BLB correction, will also effectively mitigate biases in the retrieved refractivity profiles and further-derived

atmospheric profiles. However, given the highly variable refractivity fluctuations affecting individual RO events in reality, which implies a complex dependence of the bending angle BLB on the location and the data characteristics of individual RO profiles, we found it needed to implement a BLB model with a very flexible functional behavior in order to reliably serve its purpose.



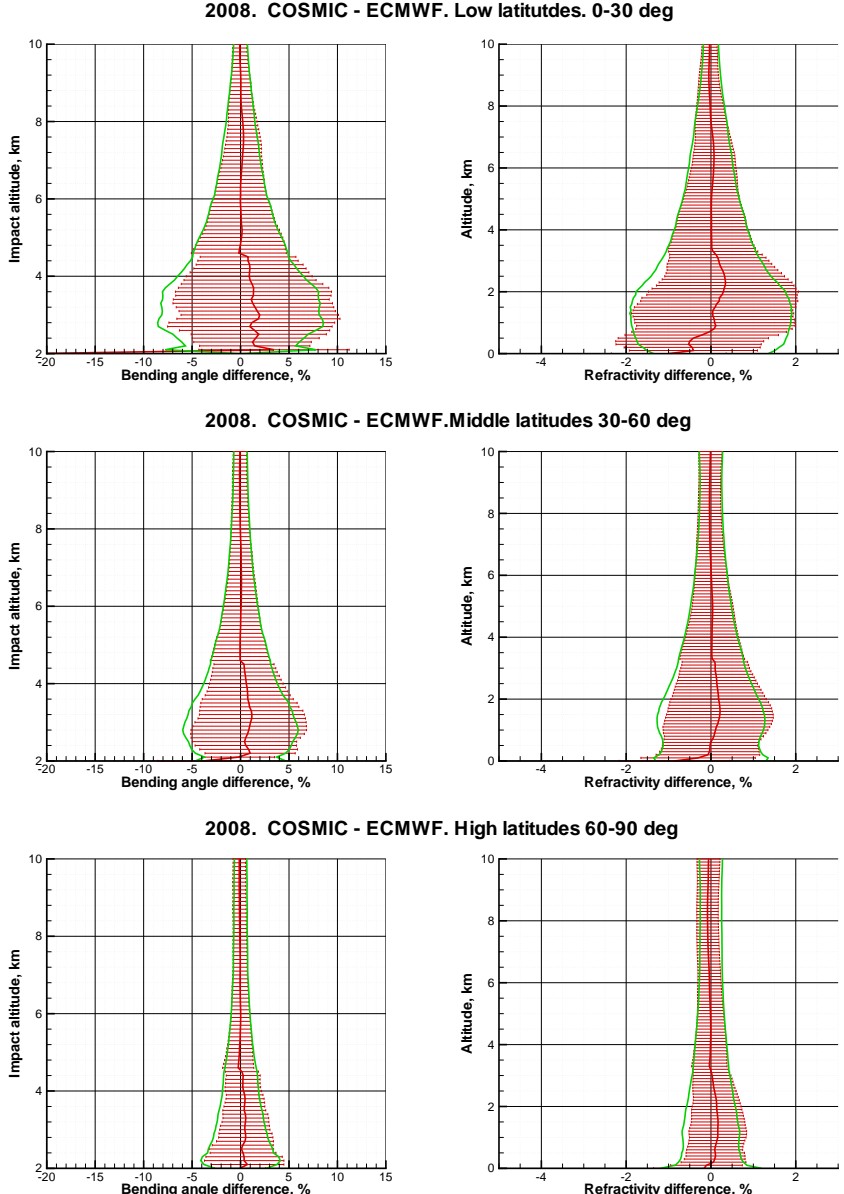

**Figure 5.** Deviation statistics based on original BLB-corrected bending angles: difference statistics of COSMIC profiles relative to ECWMF reference profiles, with same layout of panels as for Fig. 2, for bending angle as function of impact altitude (left column) and refractivity as function of altitude (right column). Results for low latitudes (top), mid latitudes (middle), and high latitudes (bottom) are shown, based on COSMIC data from the 17th day of every month of year 2008 and concurrent ECMWF analysis fields.

We therefore have chosen a versatile empirical regression modeling approach and found suitable predictors of the BLB in lower tropospheric bending angle, including: bending angle and its standard deviation, CT amplitude and its fluctuation index,





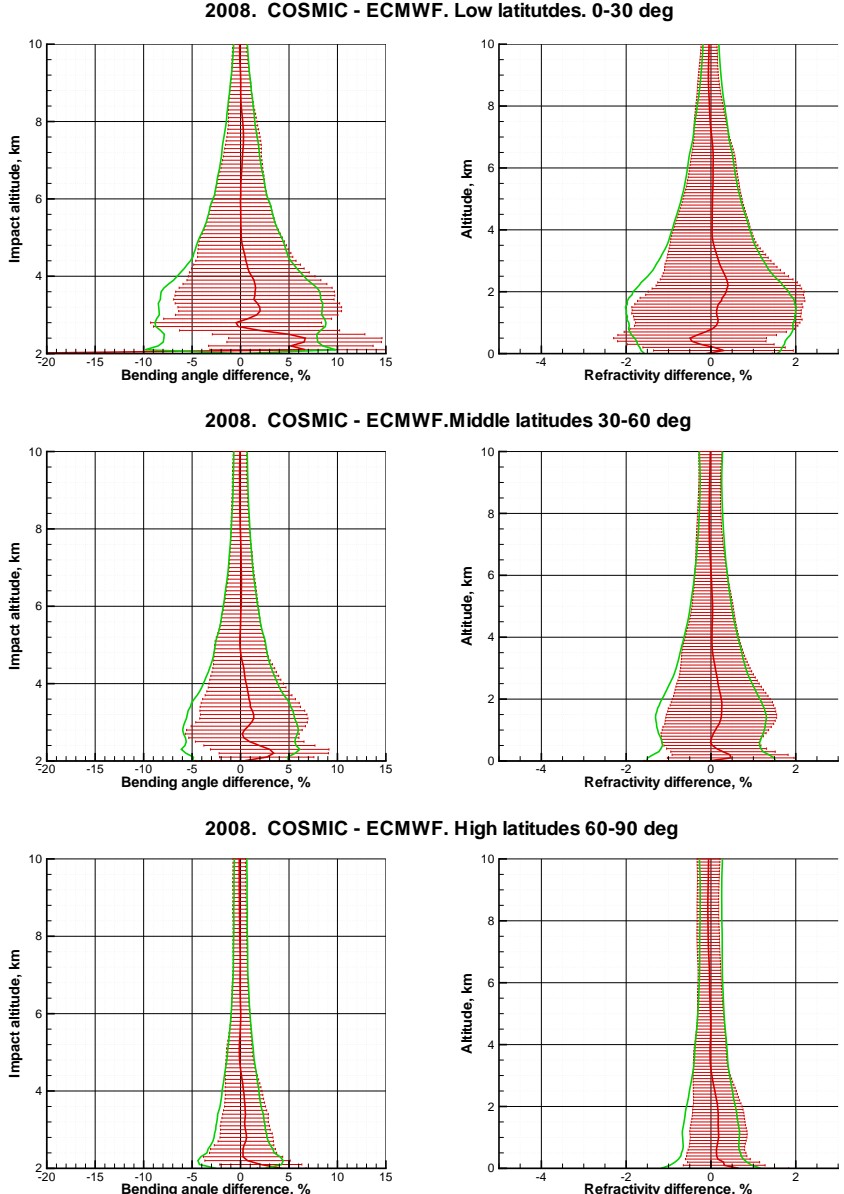

**Figure 6.** Deviation statistics based on BLB-corrected retrieved bending angles (after back-projection of original BLB-corrected bending angles to excess phases and in turn retrieving the bending angles again): COSMIC-ECMWF difference statistics with the same layout and using the same COSMIC and ECMWF data as for Fig. 5.

impact altitude and its trigonometric functions, and trigonometric functions of latitude. Degrees and cross-products of these predictors were used to form a set of flexible adaptive functions that served as basis for the BLB model, which was then obtained by regression to a large ensemble of COSMIC and ECMWF profile differences. Also a simple (residual) systematic



uncertainty model was formulated, applying to the bending angles after BLB correction. For any given RO event, the BLB model profile can be computed based on the predictors that purely depend on the event location and the characteristics of the bending angle and CT amplitude profiles.

Together with the linearized wave-optics (random) uncertainty propagation approach described by Gorbunov and Kirchengast (2015) we used the new approach to formulate the algorithmic sequence for wave-optical retrieval of bending angles from excess phases including consistent BLB correction and associated random and systematic unceratainty propagation. Evaluating the consistency of the BLB-corrected bending angles and their asssociated retrieved refractivities we achieved a successful basic validation of the new procedure: we found that the BLB correction delivers a substantial decrease of the boundary layer biases, by about a factor of five, consistent with our initial model of residual systematic uncertainty.

These results are encouraging for follow-on work in the near future that can provide a refined BLB model design and a detailed inspection and validation of the complete wave-optical retrieval and uncertainty propoagation as introduced in this study. In this way, the rOPS geometric-optical bending angle retrievals (Schwarz et al., 2017b), generally available reliably from the middle troposphere upwards, can be complemented and merged, from the upper troposphere downwards, with these wave-optical bending angle retrievals. Jointly this provides high quality of the RO data and their integrated uncertainty estimates from the stratosphere down close to the surface.

## 6   Code availability

The code used in this study does not belong to the public domain and cannot be distributed.

## 7   Data availability

COSMIC radio occultation data are freely available. To get access to them, it is necessary to sign up at the website of the CDAAC: http://cdaac-www.cosmic.ucar.edu/cdaac/ (follow the "Sign up" link for further details). ECMWF analyses are not free products and can only be obtained subject to licensing conditions depending on country and other factors. Information about ECMWF datasets and availability from the archive is provided at http://www.ecmwf.int/en/forecasts/accessing-forecasts; the commercial catalogue can be found at http://www.ecmwf.int/en/forecasts/datasets/catalogue-ecmwf-real-time-products.

*Author contributions.* Both authors formulated the initial approach of integrating wave-optical uncertainty propagation into the reference occultation processing system (rOPS) and the overall study design. Michael Gorbunov conceived and developed the bias modeling approach and the boundary layer bias model, performed the computational work and the analysis, prepared the figures, and wrote the first draft of the manscript. Gottfried Kirchengast provided input on the bias and uncertainty modeling design and feedback during the work and significantly contributed to the writing of the manuscript. Both authors contributed to consolidating the manuscript for submission and publication.



*Competing interests.* The authors declare that they have no conflicts of interest.

*Acknowledgements.* Work on Sections 1 and 2 was supported by the Russian Foundation for Basic Research (grant No. 16-05-00358-a). Work on Sections 3 and 4 was supported by the Austrian Research Promotion Agency FFG within the Austrian Space Applications Programme ASAP (ASAP-9 project OPSCLIMPROP and ASAP-10 project OPSCLIMTRACE). We acknowledge Taiwan's National Space Organization (NSPO) and the University Corporation for Atmospheric Research (UCAR) for providing the COSMIC RO data via the COSMIC Data Analysis and Archiving Center (CDAAC). We acknowledge the European Centre for Medium-Range Weather Forecasts (ECMWF) for providing the atmospheric analysis fields.



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
