# Peer review of "Wave-optics uncertainty propagation and regression-based bias model in GNSS radio occultation bending angle retrievals"

_Atmospheric Measurement Techniques, 2017_

## Referee Comment (RC1) · Anonymous Referee #1 · 15 Jun 2017

The authors present a new method for bias correcting GPS radio occultation (GPS-RO) measurements in the boundary layer. The boundary layer bias (BLB) model corrects a negative bending angle and refractivity bias in the boundary layer. The source of this bias is thought to be small scale fluctuations in the refractivity field. These are included in some simulations to demonstrate the effect. It is shown that the new bias model removes most of the BLB.

Given that one reasons GPS-RO has had an impact in operational NWP and climate re-analyses is that it can be assimilated without bias correction, introducing a bias correction scheme for this data is a significant step. I believe more is required to demonstrate

that the small scale fluctuations are the *main source of the bias*, and some discussion of the operational monitoring statistics from the NWP centres is also required.

The following questions should be considered before publication.

Is the proposed bias correction model aimed at NWP or other applications?

Please provide spatial maps of the bending angle and refractivity biases, related to the profile information shown in Figures 1 and 2. How do the spatial maps of the simulated data in Figure 1 compare with "observed" COSMIC minus ECMWF bias maps? How do the observed bias maps correlate with parameters such as low cloud cover, and total column water? Can we be sure that the small-scale fluctuations are the main source of the bias?

Figure 2. This is not consistent with standard operational monitoring at NWP centres. See, for example. http://www.romsaf.org/monitoring/index.php.

EG, GRAS measurements are biased positive with respect to both the ECMWF and Met Office models in the lower troposphere, and there are also differences for rising and setting data. This issue is complicated because the forward models used in the NWP systems compute bending angles, but use a maximum gradient ($\sim$half ducting) in their computations for numerical reasons. Furthermore, they do not compute bending angles below ducting levels. Are similar restrictions used here? The point being that a relatively simple change like this, can have a significant impact on the sampling and subsequent biases, even changing the sign of the bias.

It seems that the bias correction model presented here would currently make Met Office and ECMWF bending angle biases worse. Is that correct? Should the model be applicable to GRAS data?

The bias correction model given in section 2 seems overly complicated and requires many predictors. How many predictors are used typically in the radiance bias correction schemes? Are more predictors required here? Why?

---

## Author Comment (AC1)

Response to reviews of paper "Wave-optics uncertainty propagation and regression-based bias model in GNSS radio occultation bending angle retrievals" by M.E. Gorbunov and G. Kirchengast

Anonymous Referee #1

*The authors present a new method for bias correcting GPS radio occultation (GPS-RO) measurements in the boundary layer. The boundary layer bias (BLB) model corrects a negative bending angle and refractivity bias in the boundary layer. The source of this bias is thought to be small scale fluctuations in the refractivity field. These are included in some simulations to demonstrate the effect. It is shown that the new bias model removes most of the BLB.*
*Given that one reasons GPS-RO has had an impact in operational NWP and climate re- analyses is that it can be assimilated without bias correction, introducing a bias correction scheme for this data is a significant step. I believe more is required to demonstrate that the small scale fluctuations are the **main source of the bias**, and some discussion of the operational monitoring statistics from the NWP centres is also required.*

In fact, we do not believe small-scale fluctuations to be the main source of the bias. The small-scale fluctuation model is only used as a convenient structural model that allows finding objective candidates for bias predictors. The presented patterns in the scatter plots for the simulated and the real data indicate that the reality is more complicated than this model based on just one factor. We modify the corresponding text in the Introduction as follows:

"Although this model cannot be looked at as a complete explanation of the bias, is serves as a convenient structural model that allow exposing probable candidates for the role the objective characteristics of the signal received that may correlate with the bias. These characteristics will hereafter be referred to as predictors in the BLB model."

*The following questions should be considered before publication.*
*Is the proposed bias correction model aimed at NWP or other applications?*

We do this for an RO data processing chain with integrated uncertainty propagation where we follow the rules of the GUM,…; applications of the retrieved data are more cal/val of other observing systems, atmospheric process studies, climate applications. (And if you want to be provocative: if NWP guys use standard 1D/2D Abelian transform-/Simple raytracing-based forward operators, also they are better out with using our bias-corrected bending angle data rather than uncorrected ones, see above;)

*Please provide spatial maps of the bending angle and refractivity biases, related to the profile information shown in Figures 1 and 2. How do the spatial maps of the simulated data in Figure 1 compare with "observed" COSMIC minus ECMWF bias maps? How do the observed bias maps correlate with parameters such as low cloud cover, and total column water? Can we be sure that the small-scale fluctuations are the main source of the bias?*

We do not believe the small fluctuations to be the main source of the bias. Moreover, the proposed bias model does not use this model, as explained above. We provided spatial maps of the bias, Figures 3 and 8 (bias before and after application of our bias correction procedure). Discussion of the bias correlation with low cloud cover and total column water is beyond the scope of our paper, because our procedure is only based on predictors that derived from the objective characteristics of the signal received and observation geometry.

*Figure 2. This is not consistent with standard operational monitoring at NWP centres. See, for example.*

*EG, GRAS measurements are biased positive with respect to both the ECMWF and Met Office models in the lower troposphere, and there are also differences for rising and setting data. This issue is complicated*

*because the forward models used in the NWP systems compute bending angles, but use a maximum gradient (~half ducting) in their computations for numerical reasons. Furthermore, they do not compute bending angles below ducting levels. Are similar restrictions used here? The point being that a relatively simple change like this, can have a significant impact on the sampling and subsequent biases, even changing the sign of the bias.*

In our paper, we do not discuss nor use any special cut-off procedures used in the forward model. Note, we do not discuss any forward modeling here. On the other hand, we use the OCC package, which served as a prototype for the core ROPP package modules for RO data processing (wave optical and geometric optical inversion, statistical optimization, Abel inversion, dry temperature retrieval). These two packages were tested and found consistent. Our previous extensive study (Gorbunov et al., 2011) indicated that our results, including the negative bias of the refractivity retrievals, are in a good agreement with UCAR processing.

M. E. Gorbunov, A. V. Shmakov, S. S. Leroy, and K. B. Lauritsen, COSMIC radio occultation processing: Cross-center comparison and validation, Journal of Atmospheric and Oceanic Technology, 2011, V. 28, No. 6, 737–751, doi: 10.1175/2011JTECHA1489.1

*It seems that the bias correction model presented here would currently make Met Office and ECMWF bending angle biases worse. Is that correct? Should the model be applicable to GRAS data?*

This statement can only be made if we assume the bias correction procedure in the forward modeling mentioned by the Reviewer above. However, our approach does not involve any forward modeling. Our model only works with RO observations.

*The bias correction model given in section 2 seems overly complicated and requires many predictors. How many predictors are used typically in the radiance bias correction schemes? Are more predictors required here? Why?*

Radiance bias correction is beyond the scope of our paper. Our bias correction model is very fast: we can hardly notice any increase in the computational time; the core Fortran module is only 574 lines long (including extensive comments); in addition there is a file with 214 regression coefficients. Can we really characterize this as *overly complicated*?

Anonymous Referee #2

*The manuscript is interesting and follows in general appropriate logic flow. It requires however some improvements, as detailed below. I recommend minor revision. I encourage otherwise the authors to work to clarify the text, as it is at times difficult to follow.*
*My main concerns are presently not addressed in the paper, but could be addressed with better justification, explanation, or reference to external material. These are the following two items:*
*1) The authors show that fluctuations following the structure function presented in Fig i produce a negative bias that is very similar to the one generally known to exist. That structure function is not unreasonable, but the authors do not present a link between known or expected atmospheric properties of turbulence, or temperature and moisture fluctuations, and the $C_N^2(z)$ presented. Why that profile of fluctuations? A later sentence (P4L9) says "refractivity fluctuations can explain and quite well describe the systematic and random error...". The agreement found actually means that some fluctuation profile can be found that reproduces the known bias, although it has not been shown or referenced whether that profile was realistic*

at all. Beyond, the $C_N^2(z)$ shown is peaked at the low troposphere, descends near the surface, and also monotonically reduces above the PBL. A realistic $C_N^2(z)$ may have also other minor peaks and features.

We agree that these points need clarification in the paper, and some of the current formulations can be misleading for a reader. As we already stated above, in the rebuttal to Reviewer #1 comments, this model is a good structural model that allows finding good candidates for the bias predictors. All the bias estimates are based on the objective characteristics of the signal received. As discussed in more detail, in the paper by Gorbunov, Vorobiev, and Lauritsen (2015), that, with the corresponding choice of the effective profile of $C_N^2(z)$, the fluctuation model can reproduce the statistical characteristics of the observed bias. However, because this model is not directly used in the bias correction procedure, the further discussion of $C_N^2(z)$ is beyond the scope of this paper. We updated the formulations in the paper along the lines of this discussion. In particular, we added the following remark to the discussion of Figures 4 and 5:
"An important conclusion from these comparisons is that the fluctuation model alone does not explain the patterns observed in the real observations. However, the role of this model is to help finding reasonable predictors. The further bias correction procedure is only based on the predictors that can be readily derived from observations, rather than on the fluctuation model."

2) Although the idea of estimating the expected bias through an atmosphere of given fluctuation properties is interesting, the proposed solution is an empirical regression, where the bias (wrt ECMWF) is reduced. I am concerned about the impact on traceability, since the lower bias is obtained by heuristic fit, rather than by a physical link. Among other concerns, it simply succeeds on reproducing the bias of ECMWF (which may itself be biased) with a large number of predictors. This is the procedure normally applied to, for instance, radiance measurements. Historically, one of the major benefits of radio occultations has been the possibility to use these data without such heuristic bias correction. Otherwise, the number of predictors and

*adaptive functions being so large, it would have been surprising not to be able to fit the bias. A bias reduction with a very small number of predictors, and more physically based, would be more solid.*

Historically, there was a belief that RO data were not biased. However, the further development indicated that these data were indeed biased. Currently, there is no a good physical model that can qualitatively describe the bias. We can only say that the observed bias is a multi-factor phenomenon. In this paper, we are discussing an empirical approach to the estimate of the bias from well-defined predictors derived from objective characteristics of measurements. It is true that still we need some reference, and if we use ECMWF data, we involve the bias immanent to ECMWF. On the other hand, the method itself will stay, if we include some independent bias estimate of ECMWF. We added some remarks along these lines to the Conclusions.

*Several minor details follow.*
*P8L8: "energy density of rays". Please define the meaning here of "energy density".*
More precise formulation is "the normalized the energy distribution over rays in the impact parameter space." For more details, see the references (Gorbunov, 2002; Gorbunov, 2004).

*P10L11: Given those many predictors, one question that arises is why this set? Why not others, such as season, topography, land/ocean?*
Season was tested and found to be a weak predictor. Topography and land/ocean may be worth further investigating, although, as suggested by the new Fig. 8, they are not unambiguous.

*P10L18: "limiting the adaptive functions to the reasonable ones" What is the meaning of "reasonable" here?*
We agree that the word "reasonable" has no precise definition in the context. We re-phrased this as follows: "… apply some additional constraints in order to reduce the number of adaptive functions."

*P14L26: "reasonable profiles of $C_N^2(z)$". It has not been justified that these are reasonable. Only that they would reproduce the bias.*
Yes, as already discussed in the previous responses.

*Figures 5 and 6: Is it my perception or the procedure is moderately overcorrecting*
To some extent, they are. However, The data processing chain with the uncertainty propagation requires the back projection of bending angle bias to the excess phase and amplitude.

---

## Author Response (AR2)

Response to reviews of paper "Wave-optics uncertainty propagation and regression-based bias model in GNSS radio occultation bending angle retrievals" by M.E. Gorbunov and G. Kirchengast

Anonymous Referee

*Figure 8 suggests that the bias correction scheme in increasing positive biases over large parts of the globe (e.g., Africa, Australia). This needs to be explained.*

We modified the comment to Figure 8 as follows:
This plot indicates that, although the overall average bias is minimized, there are some regional maxima and minima. Some of them correspond to the areas with a sharp marine boundary layer (Xie et al., 2006, 2010; Gorbunov, 2014), where the negative bias is reduced but still remains. Other regions with larger deviations are located above Northern Africa and Australia, where there is a positive over-correction. The latter regions correspond to a similar terrain type, i.e., dry desert areas. This indicates the need for refined predictors, taking into account such regional effects, in order to further mitigate in a next step these more specific biases.

*I would still argue that the reader would find information on the typical number of predictors used in a radiance bias correction scheme useful. I think this is typically 6-8.*

We added this remark and complemented it with a reference to (Zhu et al., 2014).

Editor

*Please mention briefly what the "real" data is, e.g. UCAR phase data processed at Graz (e.g. for figure 3)?*

The COSMIC data were processed by the OCC package for RO data processing, as described in Gorbunov et al. (2006). This remark has been added to the text.

*It was not clear to me if figure 3 and 8 show the same information, just before and after the bias correction (you do this link for other figures). If they do, please indicate this in the text.*

Ok, in the text paragraph referring to Figure 8, we have now added a reference also to Figure 3, indicating that Figure 8 is similar to Figure 3, but presents the bias map after the BLB correction.

*And, if they do, I was too wondering why the bias does change quite substantially at all latitudes (even high ones). Isn't that a potential issue?*

This remark was co-addressed in our modified text related to Figure 8 (see answer to first comment above). The remaining bias variations indicate the need for more refined predictors in a next step that would aim at further mitigating also several regional residual biases.

*It would be nice if some information can be given why ECMWF should be biased at high latitudes.*

As visible from Figures 3 and 8, both COSMIC–ECMWF bias and its correction are small at high(er) latitudes, i.e., latitudes exceeding about 45°. We do not see any indications that ECMWF is biased there.

**Wave-optics uncertainty propagation and regression-based bias model in GNSS radio occultation bending angle retrievals**

Michael Gorbunov[1,2] and Gottfried Kirchengast[3,4]

[1]A. M. Obukhov Institute of Atmospheric Physics, Russian Academy of Sciences, Moscow, Russia.
[2]Hydrometeorological Research Centre of Russian Federation, 123242, Moscow, B. Predtechensky per., 11-13
[3]Wegener Center for Climate and Global Change (WEGC), University of Graz, Graz, Austria.
[4]Institute for Geophysics, Astrophysics, and Meteorology/Institute of Physics, University of Graz, Graz, Austria.

*Correspondence to:* Michael Gorbunov (gorbunov@ifaran.ru)

[revised manuscript text omitted]

**2.1  Underlying Model of Refractivity Fluctuations**

In order to formulate our approach to the bending angle BLB in terms of "negative refractivity bias" (Sokolovskiy et al., 2010)

30 we use the fluctuation-based model introduced by Gorbunov et al. (2015). This model is used as a simple structural model that allows finding good candidates for the objective characterisitics of the observed signals that correlate with the bias. Figure 1 shows an example profile of the refractivity structure constant $C_N^2(z)$ and the corresponding relative difference statistics

of an ensemble of bending angle and refractivity profiles. The latter were obtained by comparison of the modeled "truth" based on ECMWF refractivity fields, used as reference, and perturbed data based on the same ECMWF fields but with random refractivity fluctuations according to the $C_N^2(z)$ profile superimposed. The $C_N^2(z)$ profile was tuned to realistically represent BLB statistics of RO observations and the wave optics propagator (WOP) package (Gorbunov, 2011) was used to realistically compute the bending angles.

It is visible in  Figure 1 that the refractivity fluctuations lead to a negative refractivity bias of up to about 2 % in the boundary layer and an associated negative BLB in bending angle of up to about 5 %, typical of biases seen in real RO data. Random differences (standard deviation) reach realistic values as well, about 1.5 % in refractivity and about 5 % in bending angle.

To put these simulation results into direct context with real data,  Figure 2 shows another set of difference statistics for bending angles and refractivities, from low latitudes to high latitudes, where we again used the modeled "truth" from ECWMF fields as reference but now to illustrate the differences of observed profiles from COSMIC. The COSMIC data were processed by the OCC package for RO data processing, as described in (Gorbunov et al., 2006). These results confirm that the refractivity fluctuations model, with corresponding settings, can reproduce the systematic and random error behavior of RO bending angles and refractivities in the boundary layer. A somewhat higher level of RMS deviations (standard deviation) seen for the COSMIC data, compared to  Figure 1, is likely caused by the fact that ECMWF fields themselves deviate from the real atmospheric state (see, e.g., the error modeling of Scherllin-Pirscher et al. (2011b, 2017)).

 Figure 3 presents a latitude-longitude map of COSMIC–ECMWF  refractivity differences at a height of 0.6 km, in terms of systematic relative refractivity deviation. These results illustrate the regional variations of refractivity bias behavior and are similar to those presented in (Xie et al., 2006, 2010; Gorbunov, 2014).

Our further strategy of the bias correction consists in the following. We preform the numerical simulation of occultation events with superimposed fluctutaions and analyze different objective characteristics of RO signals in order to find those that correlated with the simulated bias. These characteristics will be referred to as predictors. Using this set of predictors, we also compare the simulation results with the processing of real COSMIC observations. We assume that this will allow formulate the model for BLB correction, will also effectively mitigate biases in the retrieved refractivity profiles and further-derived atmospheric profiles. We have to formulate the BLB model with a flexible functional behavior in order to reliably serve its purpose.

**2.2 Bending Angle BLB Model from Regression to Adaptive Functions**

We model the BLB by a predictor-based empirical model that is flexible enough to capture the BLB behavior by suitable predictors under widely variable predictor value ranges for individual RO events. Because the dependence of the BLB model profiles from predictors is unknown *a priori*, we solve for this dependence in the form of linear combination of a set of linear and non-linear functions of the predictors. We refer to these functions as adaptive functions. The model estimate of the regression coefficients of the linear combination is based on the comparison of a large set of bending angle observations with a reference data set.

[Figure]

**Figure 1.** Deviation statistics induced by simulated refractivity fluctuations: refractivity structure constant $C_N^2(z)$ profile (left) and associated difference statistics of ECMWF profiles with and without fluctuations superposed, for bending angle as function of impact altitude (middle) and refractivity as function of altitude (right), where mean difference (red), standard deviation (green) and the difference-ensemble spread (horizontal bars at vertical levels) are shown. COSMIC event geomentry and concurrent ECWMF analysis fields from the 15th day of every month of year 2008 were used to produce the statistics.

In this study, introducing a first reliable BLB model version, the observations are from the COSMIC mission and the reference data set consists of gridded fields of meteorological variables from ECMWF. The ECMWF data have their own systematic uncertainty, which is taken into account by letting these uncertainties flow into the estimated residual systematic uncertainty of bending angle profiles after BLB correction (Sect. 2.5).

5    The BLB model is formulated as follows. We used a set of COSMIC bending angle observations, including 24 representative days from year 2008. We adopted the 15th and 16th day of every month, amounting in total to about 54000 RO events. We used the corresponding ECMWF fields as basis for obtaining the "true" reference bending angles. To this end, we employed the Wave Optics Propagator (WOP) (Gorbunov, 2011) to generate the bending angle profiles from the ECMWF refractivity fields. We then performed a regression of the differences of observed and reference bending angles in the lower troposphere with

10   respect to the chosen adaptive functions (Sect. 2.4). The adaptive functions are formulated in terms of predictors, which are evaluated from objective characteristics of every RO event, without using the reference data (Sect. 2.3). These ingredients allow for the derivation of regression coefficients, which upon their estimation complete the BLB model then ready to be applied based on predictors from a given RO event.

Because we need to derive the regression model for widely diverse BLB behavior, we start with very general regression

15   relations. Consider two series of random variables, vector $\boldsymbol{x}_i$ and scalar series $y_i$, where the lower index $i$ enumerates the

[Figure]

**Figure 2.** Deviation statistics obtained for real RO data: difference statistics of COSMIC profiles including real fluctuations relative to ECMWF profiles without fluctuations, for bending angle as function of impact altitude (left column) and refractivity as function of altitude (right column), with same style of panels as for the difference statistics in Fig. Figure 1. Results for low latitudes (top), mid latitudes (middle), and high latitudes (bottom) are shown, for COSMIC events and concurrent ECMWF analysis fields from the 15th day of every month of year 2008.

[Figure]

**Figure 3.** Deviation statistics obtained for real RO data: latitude-longitude map of  difference statistics of COSMIC observations relative to ECWMF profiles without fluctuations, for refractivity at an altitude of 0.6 km. Results 
[revised manuscript text omitted]

[Figure]

**Figure 5.** Scatter plot of the the difference of fluctuation-effected and reference bending angle profiles (x-axis), for ECMWF simulations with refractivity fluctuations superposed (left column) and COSMIC observations (right column), respectively, versus the predictor variables (y-axis) bending angle standard deviation (top), normalized CT amplitude (middle), and CT amplitude fluctuation index (bottom). The reference bending angles are from ECWMF simulations without refractivity fluctuations superposed. The same ECWMF fields and COSMIC data as for  Figure 4 were used.

predictors. The further bias correction procedure is only based on the predictors that can be readily derived from observations, rather than on the fluctuation model.

In addition to these four predictors we utilize the RO event coordinates impact altitude $z$ and latitude $\lambda$, where $z = p - R_{LC} - U_{geoid}$, with $R_{LC}$ the local radius of curvature and $U_{geoid}$ the geoid undulation applying to the event location. We use the impact altitude $z$ directly and in form of the following six trigonometric functions of $z$,

$$\sin\left(2\pi n \frac{z - z_{min}}{z_{max} - z_{min}}\right), \cos\left(2\pi n \frac{z - z_{min}}{z_{max} - z_{min}}\right), n = 1...3, \tag{14}$$

where $z_{min}$ and $z_{max}$ are the limits of impact altitude wherein the BLB profiles are evaluated (equal to 1.5 km and 4.5 km). Latitude $\lambda$ is used in form of another six trigonometric functions of $\lambda$,

$$\sin(n\lambda), \cos(n\lambda), n = 1...3. \tag{15}$$

Altogether we therefore use $N_p = 17$ predictors, including impact altitude, the four observation-derived predictors, six functions of impact altitude, and six functions of latitude. This number of predictors exceeds that in radiation correction schemes, where 6–8 ones are typically used (e.g., Zhu et al., 2014).

**2.4 Construction of the Model's Adaptive Functions**

General adaptive functions as we use here are constructed in form of different degrees of the predictors and their cross-products, from degree zero, which produces unity, up to some maximum degree $D_p$,

$$\left\{\varphi^j(\mathbf{x})\right\} = \left\{1, \quad \left(x^i\right)^\gamma, \quad \left(x^{i_1}\right)^{\gamma_1}\left(x^{i_2}\right)^{\gamma_2}\right\}, \tag{16}$$

$$\leq i \leq N_p, \quad 1 \leq \gamma \leq D_p, \tag{17}$$

[revised manuscript text omitted]

 Figure 8 presents a latitude-longitude map of COSMIC–ECMWF difference at a height of 0.6 km in terms of systematic relative refractivity deviation,  similar to Figure 3 but after the BLB correction applied to the underlying bending angles. This plot  indicates that, although the overall average bias is minimized, there are some  regional maxima and minima . Some of them correspond to the areas with a sharp marine boundary layer (Xie et al., 2006, 2010; Gorbunov, 2014), where the negative bias is reduced but still remains. Other regions with larger deviations are located above Northern Africa and Australia, where there is a positive over-correction. The latter regions correspond to a similar terrain type, i.e., dry desert areas. This indicates the need for refined predictors, taking into account such regional effects, in order to further mitigate in a next step these more specific biases.

**5  Conclusions**

In this study we developed a regression-based approach for modeling and propagating atmospheric boundary layer biases (BLBs) and associated (residual) systematic uncertainties within the wave-optical retrieval chain of the reference occultation processing system (rOPS), a new RO processing system with integrated uncertainty propagaqtion that focuses on calibration/validation and climate applications.

Currently, there is no a quantitative physical model describing BLB in RO data, although there was a series of studies discussing different mechanisms resulting in BLB. The starting point encouraging and informing our BLB model design was

[Figure]

**Figure 6.** Deviation statistics based on original BLB-corrected bending angles: difference statistics of COSMIC profiles relative to ECWMF reference profiles, with same layout of panels as for  Figure 2, for bending angle as function of impact altitude (left column) and refractivity as function of altitude (right column). Results for low latitudes (top), mid latitudes (middle), and high latitudes (bottom) are shown, based on COSMIC data from the 17th day of every month of year 2008 and concurrent ECMWF analysis fields.

fluctuation-based explanatory modeling of the well known "negative refractivity bias" problem in the boundary layer. We

[Figure]

**Figure 7.** Deviation statistics based on BLB-corrected retrieved bending angles (after back-projection of original BLB-corrected bending angles to excess phases and in turn retrieving the bending angles again): COSMIC-ECMWF difference statistics with the same layout and using the same COSMIC and ECMWF data as for Figure 6.

showed that it is possible to achieve a reasonable agreement with observed bending angle and refractivity biases by modeling fluctuation statistics consistent with reasonable tropospheric profiles of the refractivity structure constant $C_N^2(z)$.

[Figure]

**Figure 8.** Deviation statistics based on original BLB-corrected bending angles:  latitude-longitude map of  difference statistics of COSMIC observations relative to ECWMF profiles without fluctuations, for refractivity at an altitude of 0.6 km. Results are shown for COSMIC events and concurrent ECMWF analysis fields from the 1st, 11th, and 21th day of every month of year 2008.

Based on this understanding we can robustly assume that reliable modeling of the bending angle BLB, and subsequent use of the model for BLB correction, will also effectively mitigate biases in the retrieved refractivity profiles and further-derived atmospheric profiles. However, given the highly variable refractivity fluctuations affecting individual RO events in reality, which implies a complex dependence of the bending angle BLB on the location and the data characteristics of individual RO
5   profiles, we found it needed to implement a BLB model with a very flexible functional behavior in order to reliably serve its purpose.

We therefore have chosen a versatile empirical regression modeling approach and found suitable predictors of the BLB in lower tropospheric bending angle, including: bending angle and its standard deviation, CT amplitude and its fluctuation index, impact altitude and its trigonometric functions, and trigonometric functions of latitude. Degrees and cross-products of these
10   predictors were used to form a set of flexible adaptive functions that served as basis for the BLB model, which was then obtained by regression to a large ensemble of COSMIC and ECMWF profile differences. Also a simple (residual) systematic uncertainty model was formulated, applying to the bending angles after BLB correction. For any given RO event, the BLB model profile can be computed based on the predictors that purely depend on the event location and the characteristics of the bending angle and CT amplitude profiles.

Together with the linearized wave-optics (random) uncertainty propagation approach described by Gorbunov and Kirchengast (2015) we used the new approach to formulate the algorithmic sequence for wave-optical retrieval of bending angles from excess phases including consistent BLB correction and associated random and systematic unceratainty propagation. Evaluating the consistency of the BLB-corrected bending angles and their asssociated retrieved refractivities we achieved a successful basic validation of the new procedure: we found that the BLB correction delivers a substantial decrease of the boundary layer biases, by about a factor of five, consistent with our initial model of residual systematic uncertainty.

Our bias model uses ECMWF fields as a reference, therefore, it involves the biases that are imminent to ECMWF model. However, the same approach can be applied together with an independent estimate of the ECMWF biases. In this study, we assumed that ECMWF biases form a small fraction of the observed systematic COSMIC–EMCWF differences.

[revised manuscript text omitted]